# Dual-Band and Multi-State Polarization Conversion Using aTerahertz Symmetry-Breaking Metadevice

**DOI:** 10.3390/nano13212844

**Published:** 2023-10-27

**Authors:** Yuwang Deng, Qingli Zhou, Xuteng Zhang, Pujing Zhang, Wanlin Liang, Tingyin Ning, Yulei Shi, Cunlin Zhang

**Affiliations:** 1Key Laboratory of Terahertz Optoelectronics, Ministry of Education, and Beijing Advanced Innovation Center for Imaging Theory and Technology, Department of Physics, Capital Normal University, Beijing 100048, China; 15624950803@163.com (Y.D.); archimilod@126.com (X.Z.); pujzhang@163.com (P.Z.); 18311031210@163.com (W.L.); 5608@cnu.edu.cn (Y.S.); cunlin_zhang@cnu.edu.cn (C.Z.); 2Shandong Provincial Engineering and Technical Center of Light Manipulations & Shandong Provincial Key Laboratory of Optics and Photonic Device, School of Physics and Electronics, Shandong Normal University, Jinan 250358, China; ningtingyin@sdnu.edu.cn

**Keywords:** terahertz, symmetry-breaking metadevice, dual band, multi state, polarization conversion

## Abstract

We numerically and experimentally demonstrate a terahertz metadevice consisting of split-ring resonators (SRRs) present within square metallic rings. This device can function as a dual-band polarization converter by breaking the symmetry of SRRs. Under *x*-polarized incidence, the metastructure is able to convert linearly polarized (LP) light into a left-hand circular-polarized (LCP) wave. Intriguingly, under *y*-polarized incidence, frequency-dependent conversion from LP to LCP and right-hand circular-polarized (RCP) states can be achieved at different frequencies. Furthermore, reconfigurable LCP-to-LP and RCP-to-LP switching can be simulated by integrating the device with patterned graphene and changing its Fermi energy. This dual-band and multi-state polarization control provides an alternative solution to developing compact and multifunctional components in the terahertz regime.

## 1. Introduction

Terahertz (THz) technology has garnered much attention in recent years due to its emerging and promising applications in telecommunications, imaging, and spectroscopy [1,2,3]. The use of this technology requires a plethora of effort devoted to the manipulation of the THz wave. However, effective THz functional devices, such as polarization control components, are still lacking. As one of the inherent characteristics of electromagnetic waves, polarization is of great practical significance, especially in sensing, holography, and stealth [4,5,6]. Metasurfaces [7], governed by artificially designed subwavelength structures, are often exploited to serve as THz polarizers. The high demand for these outstanding metastructures is ascribed to their extraordinary properties, including the flexible and effective modulation of frequency, amplitude, phase, and polarization. Some THz metadevices like single- or multi-layer wire grids [8,9], all-dielectric metamaterials [10], and chiral metasurfaces [11] for orthogonal and linear–circular polarization conversion have been investigated. Comprehensive studies on these metadevices have mainly concentrated on converting the linearly polarized (LP) wave into single circularly polarized light or its cross-output state. Few studies have been carried out on multi-band polarization conversion using the multi-state switching of the THz wave in a single device, especially in the transmission mode. In addition, to achieve the dynamic regulation of polarization, active materials containing Si [12], VO_2_ [13], liquid crystals [14], and graphene [15] have been introduced into metastructures owing to the demand for multi-functional and miniaturized optical systems. Graphene, with its remarkable features [16,17] is a widely used tunable medium for reconfigurable polarization tailoring since it exhibits gate-controllable light–matter interactions through a Fermi-level shift. In particular, in the THz range, a large continuous modulation dominated by intraband transitions can be implemented by tuning the joint density of states electrically [18]. In addition, patterned graphene at the feature positions of THz hybrid metadevices has been merely used to dynamically manipulate the amplitude, phase, and resonance frequency, respectively [19,20].

Here, we design and fabricate a metamaterial-based dual-band and multi-state polarization converter in the THz regime, which is composed of split-ring resonators (SRRs) and square metallic rings. The simulated and measured ellipticity unveils that this metastructure is capable of converting LP light into left-hand circular-polarized (LCP) and right-hand circular-polarized (RCP) waves. These responses originate from the enhancement of the cross-polarized electric field and a decrease in the co-polarized electric field induced by the breaking of SRR’s symmetry. With the introduction of graphene micro-ribbons, this proof-of-concept hybrid configuration could work as tunable LCP-to-LP or RCP-to-LP polarization converters by adjusting the Fermi energy of patterned graphene micro-ribbons. Our proposed polarization control metadevice possesses the promising potential for the integration and miniaturization required when developing THz systems.

## 2. Structure Design and Method

The unit cells and microscope photos of our designed symmetric and asymmetric metadevices, composed of SRRs integrated with square metallic rings, are depicted in Figure 1. The corresponding geometric parameters of the structures were *p* = 88 µm, *h* = 70 µm, *w* = 72 µm, *w*_1_ = *w*_3_ = 4 µm, and *w*_2_ = 6 µm, respectively. The degree of asymmetry for the metasurface can be modified by changing the split gap *g* in SRRs. In our study, the four values of parameter *g* were set to be 0, 14, 24, and 48 µm, respectively. All the other parameters were constant. These periodic arrays were fabricated on quartz substrates with a thickness of 500 μm using standard photolithographic methods. During this fabrication, a layer of photoresist material was spin-coated on the polished surface of the aforementioned quartz substrate, and ultraviolet exposure technology was applied to form the predesigned patterns of our metastructure. A 200 nm thick layer of gold (Au) with chrome at 5 nm (Cr) was evaporated on top of the film layer using magnetron sputtering. It is worth highlighting that the thin Cr film was used as the adhesive layer to enhance the adhesion between the gold layer and the quartz substrate. Then, the Cr/Au layer developed a pattern due to the lift-off operation. The corresponding optical photos of our metasurfaces with the four geometric sizes are shown in the bottom section of Figure 1. The fabricated samples were characterized under normal incidence using a standard THz time-domain spectroscopy setup to assess the time-domain and were further transformed to obtain frequency-domain spectral responses [21]. Specifically, the 800 nm source beam delivered with a Spectra Physics regenerative amplifier is divided into two paths. The first path aims to generate a THz signal, and the second path is used to probe THz light. In our measured scheme, the duration of the 800 nm source wave was 100 fs. Furthermore, the corresponding repetition rate of this source beam was 1 kHz. The generation wave was incident on a 2 mm thick <110> ZnTe crystal. Then, a 1 mm thick <110> ZnTe crystal was used to detect the transmitted THz pulse based on the free-space electro-optic technique. Moreover, the measured ambient for the THz path was purged with dry and flowing nitrogen to avoid the influence of the absorption of water vapor from the air. In our simulations, the finite element technique was employed to obtain transmission responses. This advanced simulation method, used in our study, was operated using commercial software Comsol Multiphysics (COMSOL Multiphysics 5.5, COMSOL Inc., Stockholm, Sweden). After detailed consideration of the method settings, a unit cell was selected as the simulation and calculation domain, and periodic conditions were applied to the surrounding boundaries of the chosen unit cell. The top layer in the air was regarded as the input port for THz wave incidence. Accordingly, the bottom layer in the quartz substrate operated as the output port of incident light. Here, 200 nm thick Au was treated as a perfect electric conductor. An *x*- or *y*-polarized plane wave propagating along the *z*-direction is normally incident to the proposed metasurface.

## 3. Passive Polarization Control

To explore the polarization characteristics of our metastructures, we plotted and measured simulated, co-polarized (*t_xx_*), and cross-polarized (*t_yx_*) transmission spectra, as shown in Figure 2. Here, the subscripts *x* and *y* of *t_yx_* refer to the incident *x*-polarized THz wave and the corresponding *y*-polarized output light. As shown in Figure 2a, the *t_xx_* curve of the symmetric structure with *g* = 0 µm shows an apparent sharp dip in the frequency at 0.91 THz. This apparent and sharp resonant dip resulted from the induced Fano resonance [22,23]. The *t_yx_* component was in the quenched state, as illustrated in the cross–output curve. It can be observed that the missing structure in the SRR resonator played a key role in breaking the symmetry of our metasurface. In this case, the cross-polarized component exists in the transmission spectra of the asymmetric structure. Apparently, considerable changes in transmission responses occur based on the introduction of the symmetry-breaking feature in this metallic resonator, as shown in Figure 2b. The transmission curve shows the coexistence of two Fano resonances in the *t_xx_* component for the case of parameter *g* = 14 µm. Significantly, dual-band cross-polarization conversion frequency windows (*f*_1_ and *f*_2_) with maximum transmission values of 0.31 and 0.46 appeared at the frequencies of 0.96 and 1.33 THz, respectively. For the structure of parameter *g* = 24 µm, the *t_yx_* transmission with corresponding maximum values of 0.29 and 0.41, depicted in Figure 2c, slightly decreased. By further increasing *g* to 48 µm, Figure 2d presents how the polarization capability of our metadevice was largely restrained. The results unveil that the maximum of *t_yx_* at low-frequency *f*_1_ decreased to 0.27. At high-frequency *f*_2_, this maximum value was passively and apparently regulated to 0.13. The aforementioned transmission features reveal the highly dependent polarization responses on the given parameter *g*. As shown in Figure 2e–h, by varying the degree of symmetry breaking with different parameters *g,* the experimental spectra demonstrate good agreement with our simulated transmission results. Specifically, the transmission of the resonance dip at 0.90 THz with *g* = 0 µm was 0.21. For *g* = 14 µm, the highest dual-band *t_yx_* values were 0.27 at 0.9 THz and 0.22 at 1.28 THz, respectively. There was an apparent reduction in the high-frequency (*f*_2_) cross-polarized window. The transmission responses for these devices at *g* = 24 µm and *g* = 48 µm implicate similar polarization behaviors. The resonance frequencies slightly deviate from those obtained in simulations, which could be attributed to fabrication tolerance. The measured smoothness and weakening of the high-frequency *t_yx_* window resulted from the limited resolution of our setup. In addition, to link and reveal the correlation between parameter *g* and the polarization properties of our proposed structure, we plotted curves for the maximum of *t_yx_* and corresponding frequencies at dual-band cross-polarization conversion windows with the frequencies *f*_1_ and *f*_2_ and increased parameter *g* (not shown). To be more specific, the spectra indicate that the maximum of *t_yx_* at *f*_1_ decreased slightly from 0.31 to 0.27 while its frequency remained stable at 0.96 THz with increased *g*. At *f*_2_, this value changed from 0.46 to 0.13, and the frequency showed a red-shift property from 1.33 to 1.30 THz. Herein, we conclude that the polarization conversion susceptibility at *f*_2_ was more vulnerable to the parameter g than that at *f*_1_.

To elucidate the underlying mechanism of transmission behaviors of symmetric and asymmetric metamaterials, we illustrate the surface electric field distributions in both *x* and *y* directions at the frequencies of 0.96 THz (*f*_1_) and 1.33 THz (*f*_2_) for devices with *g* = 0 µm and *g* = 14 µm, as shown in Figure 3a,b. In the symmetric case, the intensity of *E_y_* is symmetric on the *y*-axis at both *f*_1_ and *f*_2_. The left and right parts have a phase difference of *π*, causing fully reciprocal suppression. Therefore, no *t_yx_* component exists for the symmetric metastructure, as depicted in Figure 2a. Under symmetry breaking, the *E_y_* distributions of zones (1) and (2) at *f*_1_ have equivalent intensity. Furthermore, an apparent phase difference of *π* occurs between zones (1) and (2) at the same frequency of 0.96 THz. The aforementioned surface electric distributions, together with the unique phase difference, result in the suppression of surface electric distributions between the two zones. However, the stronger electric field *E_y_* in zone (3) leads to a considerable polarization conversion component at *f*_1_. The distribution of *E_y_* in zone (4) is also responsible for the polarization conversion at *f*_2_, while the other parts are mutually inhibited. By comparison, a phase difference of *π* exists in zones (5) and (6). This phenomenon is also responsible for the suppression of surface electric distributions between zones (5) and (6) of component *E_x_* at *f*_1_. A phase difference of *π*, as well as the mutual inhibition of surface electric distributions are applied in zones (7) and (8). These distributions weaken the intensity of *E_x_* and further contribute to the reduction in *t_xx_*. The situations in zones from (9) to (14) of the *E_x_* at *f*_2_ have similar behaviors. It can be inferred that these responses originate from the enhancement of the cross-polarized electric field and a decrease in the co-polarized electric field induced by the symmetry breaking of SRR. In summary, the generation of the electric field perpendicular to the incident one is attributed to the symmetry breaking of SRR to achieve the conversion of polarization [24].

Considering the anisotropy of our proposed metastructure, we plotted the simulated *t_yy_* and *t_xy_* spectra with *g* = 14 µm under the *y*-polarized incidence, as shown in Figure 4a. Compared with the co-polarized results in Figure 2b under the *x*-polarized incidence, the low-frequency transmission dip presents a large red-shift feature in simulated *t_yy_* curves. Intriguingly, the apparent dual-band cross-polarization conversion windows that exist in *t_xy_* components are identical to that of *t_yx_* components. This performance can be verified by the distributions of the electric field under the *y*-polarized incidence (not shown). The above numerical results sufficiently agree with the measured transmission responses in Figure 4b. These observations indicate that our metadevice can be applied as an effective and dual-band polarizer for the random incident wave in the *x*- or *y*-polarized state. It is worth noting that our metadevice exhibits the anisotropy of the co-polarized transmission component and the isotropy of the cross-polarized transmission component under *x-* and *y-* direction incidences. This feature possesses a wide and prosperous application in the field of information encryption in the THz region.

It is known that the output polarization state can be regulated by the amplitude ratio and phase lag between co- and the cross-polarized components. Consequently, four Stokes parameters [25] are introduced below to express the output states.
(1)$$S_{0}={\left|{\tilde{t}}_{xx}\right|}^{2}+{\left|{\tilde{t}}_{yx}\right|}^{2},$$
(2)$$S_{1}={\left|{\tilde{t}}_{xx}\right|}^{2}-{\left|{\tilde{t}}_{yx}\right|}^{2},$$
(3)$$S_{2}=2\left|{\tilde{t}}_{xx}\right|\left|{\tilde{t}}_{yx}\right|\cos\varphi_{diff},$$
(4)$$S_{3}=2\left|{\tilde{t}}_{xx}\right|\left|{\tilde{t}}_{yx}\right|\sin\varphi_{diff},$$
where *φ_diff_* is the phase difference ($\varphi_{diff}=\varphi_{xx}-\varphi_{yx}$). After this, $\tan2\alpha =S_{2}/S_{1}$ and $\tan2\chi =S_{3}/S_{0}$ can be calculated. *α* defined above is the polarization azimuth angle. *χ* is ellipticity in relation to the output ellipse. Here, −45° ≤ *χ* ≤ 45° stands for the elliptically polarized state. Specifically, −45° means a perfect LCP THz wave. 0° represents the linear polarization output, and 45° suggests a perfect RCP state. Under the *x*-polarized incidence, ellipticity is −33° at 0.99 THz and −38° at 1.32 THz, as shown in Figure 4c. This behavior confirms that our metadevice could function as a dual-band LP-to-LCP converter. In the case of the *y*-polarized incidence, Figure 4d demonstrates that these values are −39° at 0.82 THz, 41° at 0.93 THz, and 42° at 1.32 THz, respectively. These intriguing results exhibit that the tri-band polarizer can be realized at different frequencies. The single-band LP-to-LCP conversion is located at 0.82 THz. The corresponding frequencies of dual-band LP-to-RCP conversions are 0.93 and 1.32 THz, respectively. The measured ellipticity spectra are illustrated in Figure 4e,f. Under the *x*-polarized incidence, the ellipticity is −28° at 0.96 THz and −17° at 1.24 THz. In the case of the *y*-polarized incidence, the results exhibit that these values are −31° at 0.81 THz, 37° at 0.91 THz, and 19° at 1.29 THz, respectively. These experimental curves are similar to simulated behaviors. The red-shift trends and decreased values at LP-to-LCP or LP-to-RCP conversion frequencies are ascribed to fabrication tolerance. The measured smoothness and weakening of frequency-dependent ellipticity windows are the reasons for the limited resolution of our measured setup. Based on the polarization responses in Figure 3 and Figure 4, we can deduce that our asymmetric metadevice is capable of integrating dual-band polarization conversion with multi-state polarization switching in the transmission mode. Such a multifunctional metastructure provides a solution that can promote the exploration of the ultracompact and integrated systems in the THz regime. Moreover, the complicated fabrication process of multidimensional and multilayered polarization facilities can be avoided in this ultrathin symmetry-breaking single-layer metamaterial. Our device also offers more intriguing possibilities to expand high-performance-plasmonic metadevices for the polarization control of THz wave generation.

## 4. Active Polarization Control

To realize active metasurfaces, it is necessary to incorporate tunable materials, of which graphene is considered a versatile platform because it possesses gate-controllable light–matter interactions by tuning Fermi energy. Moreover, graphene can be simply transferred onto metasurfaces as well as patterned into certain shapes at the feature positions of unit cells [19,20]. The use of reshaped graphene on specific spots of hybrid THz configurations has been studied for the tunable regulation of the amplitude, phase, and resonance frequency of THz waves, respectively. In this work, we propose and simulate an active hybrid graphene metadevice, as provided in Figure 5a. Specifically, three gold electrodes, including the source (S), drain (D), and gate (G), are fabricated on the quartz substrate. The 4 µm wide graphene bridges are patterned and placed on the asymmetric metastructure along the 14 µm wide gap of SRR (*g* = 14 µm). To bridge the gap in the asymmetric metasurface with graphene micro-ribbons, the patterned graphene in our hybrid configuration can be fabricated as described in the following steps [19,20]. First, CVD-grown graphene is transferred onto the prefabricated asymmetric metastructure. For the graphene transfer, PMMA (poly(methyl methacrylate), C2, Microchem) is used as a support layer. After transferring a large area of graphene, it is patterned using ultraviolet (UV) photolithography. After UV exposure and the development of a double-layer photoresist, the unprotected graphene portions are etched using the plasma technique. During this process, two types of photoresists should be used for different purposes. In the final step, the remaining graphene is peeled off from the top of this metadevice. The fabricated S and D electrodes can be positioned directly in contact with the graphene bridges to acquire sheet resistance and estimate the corresponding Fermi energy and surface conductivity. The third electrode, which is isolated from the graphene micro-ribbons, is treated as G to apply the bias gate voltage. Ionic liquid gel (Ion Gel) is one of the most efficient dielectric materials with a high-gate capacitance [26,27,28], and it can be coated on the top of the device to support the tunability of the Fermi energy (*E_f_*) of graphene. This *E_f_* can be significantly tailored using ion gel from about −1.5 to 2.5 eV via employing a small top gate voltage (∼10 V) [29]. Our scheme provides high surface areal contact between the ion gel and these electrodes to ensure the more uniform doping profiles of patterned graphene under electrostatic gating.

Here, the dynamical manipulation is accomplished by controlling the optical property of the graphene bridges, whose surface conductivity $\sigma_{g}$ can be expressed by the Kubo formula [30],
(5)$$\sigma_{g}=\frac{ie^{2}k_{B}T}{\pi \hbar^{2}\left(\omega +i\tau^{-1}\right)}\left\{\frac{E_{f}}{k_{B}T}+2\ln\left[1+exp\left(-\frac{E_{f}}{k_{B}T}\right)\right]\right\}+\frac{ie^{2}}{4\pi \hbar }\ln\left[\frac{2\left|E_{f}\right|-\hbar \left(\omega +i\tau^{-1}\right)}{2\left|E_{f}\right|+\hbar \left(\omega +i\tau^{-1}\right)}\right],$$
where *e* is the electronic charge, *k_B_* is the Boltzmann constant, *ħ* is the reduced Planck’s constant, *T* is the temperature fixed at 293 K, and *ω* and *τ* are the angular frequency of incident light and the relaxation time of graphene, respectively. The first term in Equation (1) refers to the contribution of the intraband transition. The second term is the contribution of the interband transition. In the THz range, the interband contribution is neglected due to the photon energy *ħω* being much smaller than two of *E_f_*. Under the condition *k_B_T*≪*E_f_*, $\sigma_{g}$ is dominated by intraband transitions and can be described using the Drude model [31],
(6)$$\sigma_{g}=\frac{ie^{2}E_{f}}{\pi \hbar^{2}\left(\omega +i\tau^{-1}\right)},$$

Clearly, Equation (6) exhibits that $\sigma_{g}$ can be tuned by *E_f_*. The graphene effective refractive index $n_{g}$ [32] is written as follows:(7)$$n_{g}=\sqrt{1+i\sigma_{g}/\omega \varepsilon_{0}t_{g}},$$
where $t_{g}$ is the effective thickness of graphene and $\varepsilon_{0}$ is the permittivity of the vacuum.

Based on the aforementioned simulation method in the section structure design and method, *τ* = 31 fs is performed to calculate $\sigma_{g}$ and $n_{g}$. To ensure that graphene can fully interact with the metastructure, we assume the patterned graphene layer is 1 nm higher than the embedded gold surface.

Figure 5b shows the calculated *t_xx_* and *t_yx_* curves by changing the Fermi energy of graphene bridges with *x*-polarized incidence. At 0 eV, according to Equation (6), the conductivity of graphene is in the minimum state. In this case, the bridge is unable to connect the 14 µm wide gap of the symmetry-breaking metasurface (*g* = 14 µm). Hence, under the incident *x*-polarized THz wave, the transmission plotted with solid and dashed black lines is nearly consistent with the results without graphene, as shown in Figure 2b. At 1.0 eV, purple spectra indicate that the two resonances in the *t_xx_* component are largely hindered, while the dual-band polarization conversion windows of the *t_yx_* component are almost completely suppressed. This confirms that the electrical connection between the 14 µm wide gap is strong enough when graphene bridges are at their high conductance states. In our design, to achieve the high Fermi level of 1 eV, the true required voltage of approximately 4 to 5 V was necessary [28,29]. Under the *y*-polarized incidence, the tunability of dual-band conversion windows for the *t_yx_* curves in Figure 5c is similar. These significant and dynamic transmission changes are due to the gradual connection of a 14 µm wide gap of SRR via tuning the Fermi energy.

In Figure 5d,e, the corresponding ellipticity curves are calculated based on the aforementioned four Stoke parameters. For the *x*-polarized incidence, the ellipticity ranges from −37° to −5° at 1.01 THz with increased Fermi energy from 0 to 1.0 eV. At 1.33 THz, this value changes from −36° to −6°. Combined with *α* and *χ*, the corresponding polarization states for 0 and 1.0 eV are given in the insets of Figure 5d, clearly confirming the conversion from the LCP to LP wave at two frequencies. Under the *y*-polarized incidence, Figure 5e shows that the LCP-to-LP conversion is achieved at 0.84 THz only by changing the Fermi energy from 0 to 0.2 eV. Apparently, this susceptibility to the LCP-to-LP conversion under the *y*-polarized incidence is more vulnerable than under the *x*-polarized incidence. Further increasing the Fermi energy to 1.0 eV, the dual-band RCP-to-LP tunability is realized at 0.94 and 1.34 THz, respectively. The aforementioned reconfigurable polarization states are also illustrated in these insets. According to these modulation results, we can conclude that our metadevice possesses the ability to convert LP light into LCP or RCP waves, as well as LCP or RCP into LP states. In conclusion, we illustrate an effective and electrically driven metadevice platform to construct active multifunctional THz polarization converters. This configuration is based on regulating an electrical connection at the feature positions of unit cells for our symmetry-breaking metasurfaces.

## 5. Conclusions

In summary, we have fabricated a THz metadevice composed of SRRs and square metallic rings. This device can enable dual-band and multi-state polarization conversion with the introduction of symmetry-breaking in SRRs. Specifically, the LP-to-LCP conversion is achieved under *x*-polarized incidence. In the case of *y*-polarized incidence, the metastructure can work as LP-to-LCP and LP-to-RCP converters at different frequencies. The dynamic LCP-to-LP and RCP-to-LP conversions are further realized by integrating the device with patterned graphene and changing its Fermi energy. Such multifunctional metadevices offer an effective method to promote the development of miniaturized and integrated components in the THz region.

## Figures and Tables

**Figure 1 nanomaterials-13-02844-f001:**
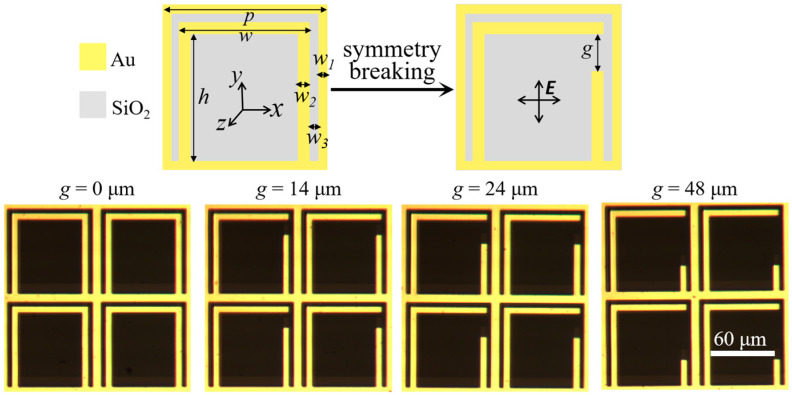
Schematic unit cells and microscope photos of fabricated metamaterials with *g* of 0, 14, 24, and 48 µm, respectively. The scale bar is 60 µm.

**Figure 2 nanomaterials-13-02844-f002:**
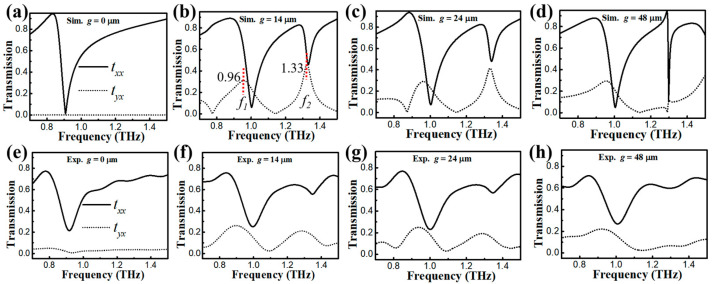
(**a**–**d**) The simulated co-polarized (*t_xx_*) and cross-polarized (*t_yx_*) transmission curves for symmetric and asymmetric metastructures with the *x*-polarized incident THz wave. (**e**–**h**) The experimental transmission spectra.

**Figure 3 nanomaterials-13-02844-f003:**
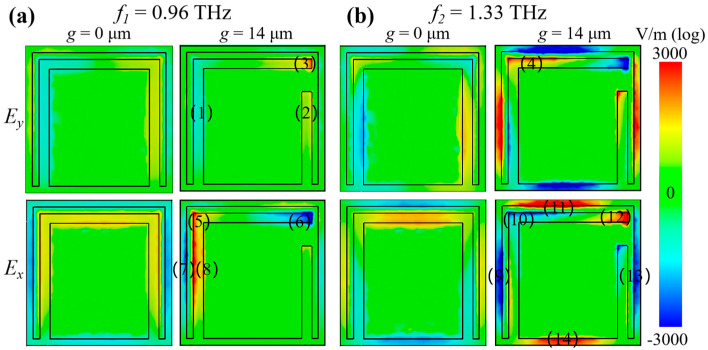
Surface *E_x_* and *E_y_* distributions at *f*_1_ (**a**) and *f*_2_ (**b**) of the symmetric (*g* = 0 µm) and asymmetric metasurfaces (*g* = 14 µm), respectively.

**Figure 4 nanomaterials-13-02844-f004:**
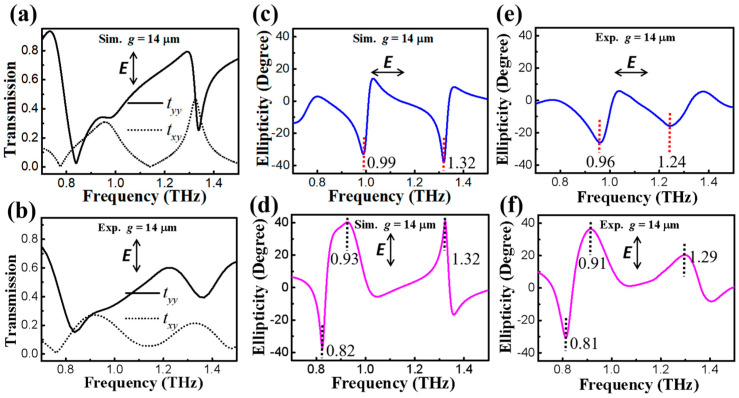
Simulated (**a**) and measured (**b**) co-polarized (*t_yy_*) and cross-polarized (*t_xy_*) transmission spectra for the asymmetric metastructure (*g* = 14 µm) with a *y*-polarized incidence. The corresponding simulated (**c**,**d**) and measured (**e**,**f**) ellipticity of the output wave with *x*- and *y*-polarized incidence, respectively.

**Figure 5 nanomaterials-13-02844-f005:**
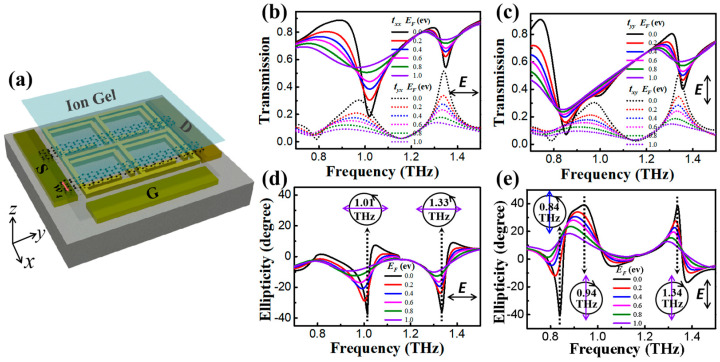
(**a**) Schematic of the hybrid metastructure with graphene (*w*_4_ = 4 µm). Simulated co- and cross-polarized transmission spectra for the hybrid asymmetric metadevice (*g* = 14 µm) against the different Fermi energy of graphene with *x*-polarized (**b**) and *y*-polarized (**c**) incidence, respectively. (**d**,**e**) The corresponding ellipticity of the output polarization wave (insets: the switching of polarization states at given frequencies).

## Data Availability

The data that support the findings of this study are available from the corresponding author upon reasonable request.

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
