# Peer review of "Dual-Band and Multi-State Polarization Conversion Using aTerahertz Symmetry-Breaking Metadevice"

_nanomaterials, 2023, doi:10.3390/nano13212844_

Round 1
Reviewer 1 Report
Comments and Suggestions for Authors
The research is well-designed, and the results are presented well.
The text should be edited for better understanding. In particular, Section 3 should be divided into five to six paragraphs. It is for readability increase.
The equations placed in the text should be treated as usual and numbered. They are in lines 83 and 86 on page eight.
I mentioned a few remarks concerning English, as well.
Comments on the Quality of English Language
The text can be understood well. It is clear what the authors want to say. However, a lot can be improved concerning the prepositions. Therefore, the text should be corrected by the professional.
Reviewer 2 Report
Comments and Suggestions for Authors
The author's deal with the standard electromagnetic problem.
What are the advantages of the proposed structure than previous.
what is the role of the lossed structures?
Is there some formula to link the parameter g to the properties of the proposed structure?
Reviewer 3 Report
Comments and Suggestions for Authors
The Authors present a periodic metasurface with square SRRs for linear to circular polarization conversion in the THz band (around 1 THz).
For the first part of their work, they demonstrate both through simulations and measurements that indeed their design acts as a dual band polarization converter and somewhat explain its operation principle. This part of the article is very good, interesting, and looks scientifically sound. The measurement configurations as well as the samples used could have been described in more detail. Also, an analysis regarding how their metasurface behaves when illuminated under oblique incidence could have been an additional asset for the paper.
In the second part of their work now, the Authors examine the same configuration but with the additional introduction of graphene which allows to tune the response via static voltage gating. The results here are only computational and look much more immature. I am listing below some comments and questions that should be addressed to improve this part of the paper.
1. The second part of the paper (after line 170) should have been a new section.
2. A separate figure with the new configuration enhanced with graphene should be included. The inset in Fig. 5 is small and has a very bad quality so the new geometry is not understandable.
3. I couldn’t understand exactly the configuration used for graphene placement, electrodes placement, graphene electromagnetic properties, and the simulation approach followed (I am referring to the text roughly between lines 175 and 189). A more thorough description of the simulation method, graphene placement, graphene properties, etc, should be included and be in general more comprehensive.
4. The Authors mention how graphene will be tuned (electrodes placement) but, in their simulations, they only state the required Fermi level to achieve such tunability. What are the true voltage levels that are necessary to achieve the high Fermi level of 1 eV in their design?
5. Reading the last paragraph, between lines 190 and 212, I was not able to understand how actually the metasurface works when Fermi level is increased. Does the polarization conversion seize? Something else? The Authors should describe this key finding of their work in more detail.
To conclude, I think that a major revision of this work is in order. The Authors should put their effort into significantly clarifying the second part of the paper.
Comments on the Quality of English Language
English are good. Some minor typos here and there so only a good proof-reading is needed.
Round 2
Reviewer 3 Report
Comments and Suggestions for Authors
The Authors responded adequately to my comments and criticism. I can recommend publication of the Article.